# Genetic Channelization Mechanism of Four Chalcone Isomerase Homologous Genes for Synergistic Resistance to Fusarium wilt in *Gossypium barbadense* L.

**DOI:** 10.3390/ijms241914775

**Published:** 2023-09-30

**Authors:** Qianli Zu, Xiaojuan Deng, Yanying Qu, Xunji Chen, Yongsheng Cai, Caoyue Wang, Ying Li, Qin Chen, Kai Zheng, Xiaodong Liu, Quanjia Chen

**Affiliations:** 1College of Agronomy, Xinjiang Agricultural University, 311 Nongda East Road, Urumqi 830052, China; xjzuqianli@126.com (Q.Z.); dengxj007@163.com (X.D.); xjyyq5322@126.com (Y.Q.); cys0620@126.com (Y.C.); wangchaoyue0810@126.com (C.W.); 18331290824@139.com (Y.L.); cqq0777@163.com (Q.C.); zhengkai555@126.com (K.Z.); 2The State Key Laboratory of Genetic Improvement and Germplasm Innovation of Crop Resistance in Arid Desert Regions (Preparation), No. 403, Nanchang Road, Urumqi 830052, China; chenxj713@163.com; 3College of Life Science, Xinjiang Agricultural University, 311 Nongda East Road, Urumqi 830052, China; xiaodongliu75@aliyun.com

**Keywords:** cotton chalcone isomerase, Fusarium wilt resistance, flavonoids

## Abstract

Duplication events occur very frequently during plant evolution. The genes in the duplicated pathway or network can evolve new functions through neofunctionalization and subfunctionalization. Flavonoids are secondary metabolites involved in plant development and defense. Our previous transcriptomic analysis of F6 recombinant inbred lines (RILs) and the parent lines after *Fusarium oxysporum* f. sp. *vasinfectum* (*Fov*) infection showed that *CHI* genes have important functions in cotton. However, there are few reports on the possible neofunctionalization differences of *CHI* family paralogous genes involved in Fusarium wilt resistance in cotton. In this study, the resistance to Fusarium wilt, expression of metabolic pathway-related genes, metabolite content, endogenous hormone content, reactive oxygen species (ROS) content and subcellular localization of four paralogous *CHI* family genes in cotton were investigated. The results show that the four paralogous *CHI* family genes may play a synergistic role in Fusarium wilt resistance. These results revealed a genetic channelization mechanism that can regulate the metabolic flux homeostasis of flavonoids under the mediation of endogenous salicylic acid (SA) and methyl jasmonate (MeJA) via the four paralogous *CHI* genes, thereby achieving disease resistance. Our study provides a theoretical basis for studying the evolutionary patterns of homologous plant genes and using homologous genes for molecular breeding.

## 1. Introduction

Cotton is an economically important crop in China, and cotton production accounts for the largest proportion of natural textile fiber production [1]. *Gossypium barbadense* plants are highly susceptible to Fusarium wilt, the occurrence of which leads to a sharp drop in cotton yield, the deterioration of fiber quality, and severe production failure. Therefore, cotton Fusarium wilt is referred to as cotton “cancer” [2]. Fusarium wilt is caused by *Fusarium oxysporum* f. sp. *vasinfectum* (*Fov*), a global soil-borne pathogenic fungus that can infect more than 100 kinds of plants [3].

To respond to infection by various pathogens, plants have evolved two main innate immune mechanisms [4]: pathogen-associated molecular pattern (PAMP)-triggered immunity (PT1) and effector-triggered immunity (ET1) [5,6]. Pathogens can complete invasion and colonization processes in the host by secreting effector molecules to inhibit the defense signaling pathway [7] or by directly entering the host [8]. In addition, terrestrial plants have evolved many specific metabolites, including a wide variety of flavonoids, to respond to various biotic and abiotic stresses [9]. Relevant studies have shown that the chalcone isomerase (*CHI*) gene plays an important role in plant disease resistance. In soybean, the overexpression of the *CHI* gene enhanced resistance to *Phytophthora sojae* [10]. In cotton, *CHI* genes have been predicted to be regulated by miRNAs in response to pathogen challenge [11]. In addition, the GbCHI protein inhibits spore germination and mycelial growth of *Verticillium dahliae* [12].

Studies have shown that flavonoids are secondary metabolites involved in plant development and defense. They have received extensive attention due to their antioxidant, free radical scavenging, and antibacterial functions, which are of great significance to human health [13]. CHI is ubiquitous in higher plants and is the second most important enzyme in the flavonoid synthesis pathway. Chalcone in plants is converted to dihydroflavones through the action of CHI, and then further converted to various flavanones and flavonoids. The *CHI* gene family can be divided into four categories [14]. Type I CHI proteins can use only 6-hydroxychalcone (naringenin chalcone) as a substrate in vascular plants to generate 5-hydroxyflavanone (2s-naringenin); type II CHI proteins not only have the catalytic function of type I proteins, but can also use 6′-deoxychalcone as a substrate to generate 5-deoxyflavanone (liquiritigenin). Moreover, the type II proteins are found only in legumes [15]. Type III CHI proteins localize to plastids and are the sites of new fatty acid (FA) biosynthesis in plant cells. In vivo, the expression profile of type III CHI proteins is associated with core FA biosynthetic enzymes. In vitro, these proteins are FA-binding proteins (FAPs) [16]. Notably, FAPs represent a specific period in plant evolution, that is, the adaptive evolution of a nonenzymatic ancestor to a stereospecific and catalytically perfected enzyme [17]. Type IV CHI proteins are found only in terrestrial plants and lack the conventional CHI activity. Previously, it was found that the CHI-like protein EFP is a type IV CHI and is involved in the early stage of the flavonoid biosynthesis pathway of Japanese morning glory, playing a role in ensuring the production of flavonoids. Its function is conserved in different terrestrial plant species, and it is a catalyst for the synthesis of flavonoids and anthocyanins [18,19,20]. Therefore, the classic type I protein and the legume-specific type II CHI are truly catalytically active CHIs that can efficiently produce naringenin.

Gene duplication is one of the core causes for the existence of multiple gene copies. Gene duplication can lead to the formation of families with new functions that may contribute to speciation [21,22,23,24], and the types of gene duplication include whole-genome duplication (WGD), tandem and segmental duplication, transposon-mediated duplication, and reverse transcription-based replication [23,25]. The most common evolutionary model is that one of the repeats retains the function of the original gene, while the other repeat is a pseudogene [25,26]. Alternatively, both copies coexist and slowly evolve over a long period, which provides functional novelty in genetic resources for the evolution of species, that is, neofunctionalization [22].

To date, the relationship between the functional evolution of CHI family genes and Fusarium wilt in *G. barbadense* has rarely been reported. According to the identification of the cotton *CHI* gene family, there are 10 *CHI* genes in *G. barbadense*, as shown in our previous studies [27]. Furthermore, some homologous genes of the cotton *CHI* gene family may have been involved in the Fusarium wilt resistance pathway of *G*. *barbadense* during evolution. Over the course of evolution, the cotton *CHI* genes may have assumed the role of disease resistance genes in *G*. *barbadense*. Our research further clarifies the molecular specificity and function of cotton *CHI* genes and their potential application value in cotton genetic improvement. In addition, the relationships of several *CHI* homolog genes with the same function in the process of genome evolution and the basis for genetic redundancy and phenotype robustness were further clarified.

At present, there are few reports on the synergistic relationship and possible neofunctionalization differences of paralogous CHI family paralogous genes involved in Fusarium wilt resistance in cotton during evolution. Our previous transcriptomic analysis of F6 recombinant inbred lines (RILs; two resistant offspring and two susceptible offspring) and the parents after *Fov* infection showed that *CHI* genes have important functions in cotton [27]. In addition, four paralogous genes in the cotton *CHI* gene family were found to be evolutionarily involved in the resistance to Fusarium wilt in *G*. *barbadense*. Over the course of evolution, the four paralogs may have assumed the role of disease resistance genes in *G*. *barbadense* [27]. In this study, we identified the molecular specificity and function of the four paralogous genes in *G*. *barbadense* and found that these genes function synergistically. This study reveals that homologous genes are synergistically regulated in the homeostasis of secondary metabolites due to gene duplication, and the antibacterial effect of flavonoids can be exerted through the “one major, three secondary” pattern of the *GbCHI* gene family (one major gene, *GbCHI05*, and three minor genes *GbCHI01*, *GbCHI06* and *GbCHI09*). This study provides evidence for genetic redundancy and phenotypic robustness. In addition, this study offers a strategy for ensuring plant stability to maintain normal growth and development in the face of genetic and environmental changes.

## 2. Results

### 2.1. Cloning and Bioinformatics Analysis of the GbCHI01, GbCHI05, GbCHI06 and GbCHI09 Genes

The open reading frame (ORF) sequences of *GbCHI01*, *GbCHI05*, *GbCHI06* and *GbCHI09*, with sizes of 618 bp, 684 bp, 618 bp and 684 bp, respectively, were cloned (Appendix A). The proteins encoded by *GbCHI01*, *GbCHI05*, *GbCHI06* and *GbCHI09* were analyzed by the biological software program ProtParam on the ExPASy website. The four proteins were all hydrophilic proteins. The physicochemical properties of paralogs on the same chromosome were similar, while those of paralogs on different chromosomes were completely different (Appendix A). The tertiary structures of the proteins encoded by the *GbCHI01*, *GbCHI05*, *GbCHI06* and *GbCHI09* genes of *G*. *barbadense* were analyzed, and the α-helix and β-sheet structures were folded to determine the correctly folded protein structure that could perform the biological functions of the genes (Appendix A). The overall structure of the GbCHI01 and GbCHI06 proteins resembled an upside-down bouquet, while the overall structure of the GbCHI05 and GbCHI09 proteins resembled an upright bouquet. The two bouquets were similar to each other, differing by only a 180-degree horizontal rotation. Motif Scan software (https://swissmodel.expasy.org/ (accessed on 14 August 2023)) was used to analyze the motifs and domains of the GbCHI01, GbCHI05, GbCHI06, and GbCHI09 proteins. The four proteins had casein kinase II phosphorylation sites, N-myristoylation sites, protein kinase C phosphorylation sites and other functional motifs. *GbCHI01* and *GbCHI06* had cAMP- and cGMP-dependent protein kinase phosphorylation sites, tyrosine kinase phosphorylation sites and BRCA2 repeat motif functions, and *GbCHI05* and *GbCHI09* had N-glycosylation sites, an ATP/GTP-binding site motif A (P-loop) and a Big-1 (bacterial Ig-like domain 1) domain profile with primitive functions (Appendix A). The GbCHI01, GbCHI05, GbCHI06 and GbCHI09 protein sequences were aligned using multiple sequence alignment by CLUSTALW and ESPript3.0 online software, and online analysis with Swiss Model software delineated the secondary structure of the GbCHI01, GbCHI05, GbCHI06 and GbCHI09 proteins. The GbCHI01 and GbCHI06 proteins had eight α-helices and 11 β-sheets. The GbCHI05 and GbCHI09 proteins had nine α-helices and 11 β-sheets (Appendix A).

### 2.2. Subcellular Localization of the GbCHI01, GbCHI05, GbCHI06 and GbCHI09 Genes

The GbCHI01, GbCHI05, GbCHI06 and GbCHI09 proteins may be localized to the nucleus and cell membrane (Figure 1). Based on the structural and functional analysis of CHI proteins described above, it is speculated that the nucleus and cell membrane are the main sites where CHI proteins function. The paralogous genes have the same subcellular localization and may perform the same function.

### 2.3. Silencing of GbCHI01, GbCHI05, GbCHI06 and GbCHI09 Reduces Resistance to Fusarium Wilt in Cotton

The *GbCHI01*, *GbCHI05*, *GbCHI06*, and *GbCHI09* genes of *G. barbadense* 06-146 were silenced by virus-induced gene silencing (VIGS) technology, and *cloroplastos alterados 1* (*CLA1*) was silenced as a positive control. Positive control (*p*TRV2::*CLA1*) cotton plants exhibited an albino phenotype after injection (Figure 2). The silencing efficiency measurements show that the expression levels of *GbCHI01*, *GbCHI05*, *GbCHI06* and *GbCHI09* were significantly decreased (Figure 2), indicating that *GbCHI01*, *GbCHI05*, *GbCHI06* and *GbCHI09* were successfully silenced. The experimental group with silenced *GbCHI01*, *GbCHI05*, *GbCHI06* and *GbCHI09* genes (*p*TRV2::*GbCHI01*, *p*TRV2::*GbCHI05*, *p*TRV2::*GbCHI06* and *p*TRV2::*GbCHI09*) and the control group (*p*TRV2::00) were observed 28 days after infection with Fusarium wilt.

The plants of the experimental group and the control group exhibited leaf yellowing, wilting and defoliation, and the plants of the experimental group exhibited more severe yellowing and wilting than those of the control group (Figure 2). Statistical analysis of the disease index (DI) showed that the DI of the experimental group was significantly higher than that of the control group. In conclusion, silencing *GbCHI01*, *GbCHI05*, *GbCHI06* and *GbCHI09* in *G*. *barbadense* 06-146 significantly reduced the resistance of cotton seedlings to Fusarium wilt (Figure 2).

The results of the recovery test of Fusarium wilt in stem segments show that in the same parts of cotton stem segments, the control plants developed fewer Fusarium wilt stem segments than the gene-silenced plants (*p*TRV2::*GbCHI01*, *p*TRV2::*GbCHI05*, *p*TRV2::*GbCHI06* and *p*TRV2::*GbCHI09*) (Figure 3). The gene-silenced plants displayed much higher *Fov* race 7 biomass. These results indicate that the gene-silenced plants had a greater extent of cotton *Fov* infection, and the ability of the gene-silenced plants to resist cotton Fusarium wilt was weaker than that of the control plants. The ability of cotton seedlings to resist cotton Fusarium wilt decreased with the silencing of the *GbCHI01*, *GbCHI05*, *GbCHI06* and *GbCHI09* genes (Figure 3).

### 2.4. Overexpression of GbCHI05 Improves Resistance to Fusarium Wilt in Arabidopsis thaliana and Cotton

After the overexpression of the *GbCHI05* gene in T_3_ *Arabidopsis thaliana* seeds, the wild type (WT) and *chi05* mutant were cultured in a greenhouse. The wild type (WT) and *chi05* mutant began to show yellow margins and curled leaves 14 days after *Fov* infection, while the transgenic *A. thaliana* plants showed good leaf growth (Figure 4). At 21 days of infection, WT *A. thaliana* and the *chi05* mutant were found to have more severe disease than the *GbCHI05-*overexpressing strains (Figure 4). Statistical analysis of the DI showed that the DIs of the WT and *chi05* mutant were significantly higher than those of the *GbCHI05-*overexpressing strains. The results show that the *GbCHI05* gene improved the resistance of *A. thaliana* to Fusarium wilt.

The results of the Fusarium wilt recovery assay show that the control plants developed fewer Fusarium wilt-infected stem segments than the *GbCHI05* mutant *A*. *thaliana* plants (Figure 4). The *GbCHI05* mutant *A*. *thaliana* plants displayed much higher *Fov* race 7 biomass. The *GbCHI05-*overexpressing *A*. *thaliana* plants developed fewer Fusarium wilt-infected stem segments than the control plants (Figure 4). The *GbCHI05-*overexpressing *A*. *thaliana* plants displayed much higher *Fov* race 7 biomass. These results indicate that the *GbCHI05* mutant *A*. *thaliana* had a greater extent of *Fov* infection, and that the *GbCHI05* mutant *A*. *thaliana* had a weaker ability to resist Fusarium wilt than the control plants (Figure 4). The *GbCHI05*-overexpressing *A*. *thaliana* plants were less susceptible to Fusarium wilt than the control plants, exhibiting a greater ability to resist cotton Fusarium wilt (Figure 4).

After the overexpression of the *GbCHI05* gene in T_3_ 06-146 seeds, the 06-146 (WT) seeds were cultured in the greenhouse. The expression level of the *GbCHI05* gene in *GbCHI05-*overexpressing 06-146 was significantly higher than that in the recipient materials, indicating that the *GbCHI05* gene was overexpressed in 06-146 (Figure 5). After 28 days of infection, the disease in control plants was more severe than that in *GbCHI05-*overexpressing strains (Figure 5). The results indicate that cotton seedlings overexpressing the *GbCHI05* gene were resistant to cotton Fusarium wilt. The DI was statistically analyzed, and the DI of the control group was found to be higher than that of the experimental group. The *GbCHI05-*overexpressing cotton plants developed fewer Fusarium wilt-infected stem segments than the control plants (Figure 5). The *GbCHI05-*overexpressing cotton plants displayed much higher *Fov* race 7 biomass. The overexpression of the *GbCHI05* gene in 06-146 enhanced the resistance of the cotton seedlings to Fusarium wilt.

### 2.5. Detection of Genes in Flavonoid Metabolic Pathways and Genes Related to Fusarium Wilt

To study the effects of silencing the *GbCHI01*, *GbCHI05*, *GbCHI06* and *GbCHI09* genes on the genes in the flavonoid metabolic pathway, the expression levels of synthetic genes in the flavonoid metabolic pathway were determined. At the same time, the expression levels of the previously reported *GbERF-like* and *Gbar_D03G002290* genes were also measured and analyzed. The study found that when *GbCHI01* was silenced, the expression levels of *GbCHI05*, *GbCHI06*, *GbCHI09*, *GbDFR*, *GbF3*′*H*, *GbFLS*, *GbANR* and *GbANS* were upregulated to varying degrees, the expression of *GbC4H* was decreased, the expression of the *GbERF-like* gene was upregulated, and the expression of *Gbar_D03G002290* was downregulated (Figure 6). When *GbCHI05* was silenced, the expression levels of *GbCHI01*, *GbCHI06*, *GbCHI09*, *GbDFR*, *GbF3*′*H*, *GbFLS*, and *GbANS* were all downregulated to varying degrees (Figure 6). The expression levels of *GbANR* and *GbC4H* first decreased, then increased and then decreased again, and *GbERF-like* expression was upregulated. *Gbar_D03G002290* expression was downregulated first and then upregulated. When *GbCHI06* was silenced, the expression levels of *GbCHI06*, *GbCHI09*, *GbDFR*, *GbF3*′*H* and *GbFLS* were upregulated to different degrees, and the expression levels of *GbCHI01*, *GbANR*, *GbANS*, *GbC4H*, *GbERF-like* and *Gbar_D03G002290* were all downregulated to different degrees (Figure 6). When the *GbCHI09* gene was silenced, the expression levels of *GbCHI01*, *GbCHI05*, *GbCHI06*, *GbDFR*, *GbANS*, *GbC4H*, *GbERF-like* and *Gbar_D03G002290* were all downregulated to varying degrees, and the expression levels of *GbF3*′*H*, *GbFLS and GbANR* were upregulated to varying degrees (Figure 6). The results show that the four homologous genes had similar regulatory genes under the conditions of *Fov* infection, but the differences in regulation were unique to each gene involved in the regulation of flavonoid metabolic flux.

### 2.6. Flavonoid Content after Silencing of GbCHI01, GbCHI05, GbCHI06 and GbCHI09 and GbCHI05 Overexpression

The following linear regression equation was established according to the standard curve of rutin (Appendix A): y = 0.1322x + 0.0968, R² = 0.9925. The flavonoid content was calculated as follows: (A508 − 0.0968)/0.1322 × 100. Statistical analysis of the flavonoid content showed that the flavonoid content in control plants was higher than that in experimental plants without *Fov* infection, and the flavonoid content in control plants was lower than that in experimental plants without *Fov* infection (Figure 7). The study results show that the *GbCHI01*, *GbCHI05*, *GbCHI06* and *GbCHI09* genes are involved in the synthesis of flavonoids, but they are not the only genes that control the synthesis of flavonoids (Figure 7). The synthesis of flavonoids is regulated by multiple genes. After *GbCHI05* gene overexpression, the levels of flavanones in *GbCHI05-*overexpressing cotton seedlings were higher than those in the recipient material 06-146 (Appendix A).

### 2.7. Levels of Endogenous Hormones and ROS in GbCHI01, GbCHI05, GbCHI06 and GbCHI09 Gene-Silenced Plants after Fov Infection

The levels of endogenous MeJA and SA in the samples were determined by ELISA, and it was found that with the prolongation of infection time, the levels of MeJA and SA increased in both the control group and experimental group (Figure 8). In addition, in the *GbCHI01*, *GbCHI06* and *GbCHI09* gene-silenced plants, the endogenous MeJA content in the samples of the experimental group was lower than that in the samples of the control group, while the endogenous SA content in the samples of the experimental group was lower than that in the samples of the control group (Figure 8). In *GbCHI05* gene-silenced plants, the endogenous MeJA content in the experimental group samples was lower than that in the control group samples. However, the amount of endogenous SA in the experimental group samples was lower than that in the control group samples, except at 2 h and 4 h (Figure 8).

The levels of ROS in the samples were determined by ELISA. The ROS content in the test group was lower than that in the control group in *GbCHI01* and *GbCHI05* gene-silenced plants. In *GbCHI06* and *GbCHI09* gene-silenced plants, the ROS content in the experimental group was higher than that in the control group (Figure 9). In *GbCHI01* and *GbCHI06* gene-silenced plants, the ROS content in the samples before infection was higher than that in the samples after infection. In *GbCHI05* and *GbCHI09* gene-silenced plants, the ROS content in the samples before infection was lower than that in the samples after infection (Figure 9).

## 3. Discussion

During plant development and evolution, genes control biological processes through precise spatial and temporal expression. Genes regulate biological processes and respond to environmental changes through changes in expression. Therefore, evolutionary adaptation and changes in genes are of great significance for the ability of individual plants to adapt to environmental changes over tens of millions of years [28,29]. The copy number changes caused by gene duplication and evolution and the differences in copy structure and function are important contributors to the construction and improvement of gene regulatory networks [30,31,32,33,34]. In addition, gene duplication leading to redundancy in gene segments may help species acquire adaptation and genetic resistance to changes in environmental conditions [35].

### 3.1. Effect of Gene Structural Variations on the Neofunctionalization Differentiation of Homologous Genes and the Synergistic Functions of Homologous Genes

Different genomic structural variations (insertions or deletions) can lead to the reorganization of topologically associated domains (TADs). Cotton species-specific TAD boundary formation occurs more frequently in inactive chromatin regions, which is related to cotton species-specific transposon insertion. At the same time, the chromatin interactions between genes are more conserved than those between noncoding regulatory regions and genes among different cotton species. It was found that structural variation, transposon insertion and changes in chromatin openness drove the evolution of regulatory networks between noncoding regulatory elements and homologous genes, providing new ideas for the study of structural variation and function in cotton [36].

We investigated the effect of gene structural variation on the neofunctionalization of homologous genes and the synergistic function of homologous genes during *GbCHI* gene evolution, and found that the sequences of paralogous *CHI* genes in different chromosomal subgroups were highly conserved, with only a few base changes. In addition, the three-dimensional structure prediction showed that the proteins encoded by the *GbCHI01*, *GbCHI05*, *GbCHI06* and *GbCHI09* genes were similar in spatial structure (Appendix A). The protein sequences of GbCHI01, GbCHI05, GbCHI06 and GbCHI09 have different amino acids at individual sites (Appendix A), which may lead to differences in their secondary structures, leading to differences in activation or enzymatic reaction rates (Appendix A).

As the main executors of gene functions, proteins need to be located in the correct subcellular compartments to function normally, maintain orderly and efficient progression of various complex biochemical processes in the organism and ensure the normal life activities of the organism [37]. Therefore, protein subcellular localization is closely related to protein function, which is the basis for studying the biological function of proteins and has important significance for proteomics [38]. The function of a gene and the metabolic pathways in which it participates are largely related to the location of the protein encoded by the gene in the organelle. Studying the subcellular localization of a gene can enable identification of the specific location of protein expression in the cell, which allows speculation around its function and the metabolic pathways involved [39]. In this study, we found that the nucleus and cell membrane might be the main functional sites of the CHI protein, which was similar to the localization of CHI proteins in *Ipomoea batatas* (Linn.) Lam. [40] and *Vitis vinifera* L. [41], indicating that CHI may play essential roles as an enzyme and in secondary metabolite synthesis during cell reproduction (Figure 1). Research on the subcellular localization of the protein encoded by the *CHI* gene provides a theoretical basis for further analysis of the function of the gene and screening of its interacting proteins.

### 3.2. Relationship between the Neofunctionalization of Homologous Genes and Flavonoid Metabolic Pathway Genes

Both whole-genome and small-scale duplications are referred to as gene duplications and are ubiquitous in plant genomes [25,42,43,44,45]. Many redundant repeat segments are generated after gene duplication, which makes genetic variation abundantly selective [25,43]. Genetic variation can be accomplished by pseudogenization, subfunctionalization or neofunctionalization [25,43,46,47]. In addition, paralog evolution by “active compensation”, in which one or more paralogs in the genome are upregulated by transcription to replace the impaired activity of other paralogs, manifests in the genome in the form of redundancy [48,49,50]. This approach offers the possibility for species to maintain stability during genetic changes and under changes in the external environment, and allows genes to remain stably inherited under selection pressure [51,52]. However, although duplication can provide many redundant fragments that allow species to cope with impaired gene activity, it also promotes genetic variation [51,53,54]. This genetic variation can occur simultaneously in the coding and regulatory region sequences of genes, and is the source of the diversity of paralogous genes. Species produce duplications for short periods, and the variation caused by this short period of duplication-induced diversity in the function of homologous genes is unknown. To date, functional studies of homologous genes have been conducted only within a single system or in a few species, and the compensatory relationships and functional changes between genes have not been systematically studied [44,48,55]. Therefore, studying the compensatory relationships of paralogous genes in Fusarium wilt resistance and their differences after neofunctionalization in *G*. *barbadense* will provide evidence for the evolutionary model after lineage gene duplication.

Our study showed that silencing *GbCHI01*, *GbCHI05*, *GbCHI06* and *GbCHI09* in *G*. *barbadense* 06-146 reduced the resistance of cotton seedlings to Fusarium wilt (Figure 2 and Figure 3). The overexpression of the *GbCHI05* gene in 06-146 and *A*. *thaliana* enhanced the resistance of cotton and *A*. *thaliana* seedlings to Fusarium wilt (Figure 4, Figure 5 and Figure 6). Therefore, the roles of these paralogous genes are not equivalent; instead, these genes work together to achieve robust genetic channelization. To study the effect of silencing the *GbCHI01*, *GbCHI05*, *GbCHI06* and *GbCHI09* genes on the genes in the flavonoid metabolic pathway, the expression levels of synthetic genes in the metabolic pathway were measured (Figure 7).

The experiment showed that under the conditions of *Fov* infection, the silencing of the homologous genes led to similar changes in the expression levels of the same genes, but the expression levels had opposite trends, and some genes exhibited different expression levels due to gene silencing. The expression patterns of *GbDFR*, *GbF3*′*H* and *GbFLS* were the same when *GbCHI01*, *GbCHI06* and *GbCHI09* were silenced, respectively. The expression patterns of *GbANS* were different when *GbCHI05* was silenced compared to when other genes were silenced. In *GbCHI01* gene-silenced plants, the expression pattern of *GbANR* was different from that in other gene-silenced plants. This indicates that the *GbCHI01*, *GbCHI05*, *GbCHI06* and *GbCHI09* genes have similar regulatory networks but may also have their own unique regulatory pathways. These genes may cooperate with and restrict each other, and jointly affect the synthesis and functions of secondary metabolites. However, the differences in their regulation may be the key to the divergence of the neofunctionalization of genes and the maintenance of gene dosage balance.

### 3.3. Relationship between Neofunctionalization of Differentiation of Homologous Genes and Flavonoid Content

Flavonoids are the largest class of polyphenols, and the initial substrates for plant flavonoid synthesis are coumaroyl-CoA and malonyl-CoA, derived from the phenylpropane metabolic pathway [56,57]. First, phenylalanine produces colorless naringenin under the action of phenylalanine ammonia-lyase (PAL), cinnamate 4-hydroxylase (C4H), 4-coumaric acid-CoA ligase (4CL), chalcone synthase (CHS) and CHI. Naringenin, as a major metabolite, enters the synthetic pathways of other flavonoids. Since the synthetic pathways of flavonoids are conserved in plants, many enzymes can change the basic skeleton of flavonoids under different external conditions, thereby generating different kinds of flavonoids [58,59].

The *CHI* gene was first isolated from French pea (*Pisum sativum*), and it has subsequently been isolated and cloned from various species [60,61,62,63,64,65,66,67]. Whether CHI is expressed in plants, and the level of expression, affect the metabolism of flavonoids in plants. Relevant studies have shown that when CHI is inactive or mutated, the chalcone content in plants increases, and the flavonoid content is significantly reduced compared with that in the presence of wild type CHI. Decreased CHI activity in carnation (*Dianthus caryophyllus* Linn.) [68] and cyclamen (*Cyclamen persicum* Mill.) [69] resulted in a change in the color of the flowers, leading to the production of yellow flowers. In addition, after the expression of the CHI gene in carnation (*D. caryophyllus* Linn.) was reduced, chalcone accumulated in large quantities, and yellow flowers were produced. Similarly, the mutation of CHI changed the flower color of carnations to yellow [70]. The mutation of the CHI gene in petunia (*Petunia hybrida Vilmorin.*) led to the formation of yellow or green pollen due to the accumulation of chalcone [60]. After the RNAi-mediated inhibition of *CHI* gene expression in tobacco (*Nicotiana tabacum* L.), the anthocyanin content in the petals decreased, the chalcone content increased, and the petals turned yellow [71]. Relevant studies have shown that the flavonoid content in plants can be significantly increased by increasing the activity of the CHI enzyme or overexpressing the associated gene. For example, the petunia CHIA gene was introduced into tomato peel, and the flavonol content was found to increase 79-fold [72]. The jellyfish snow lotus CHI gene was transferred into Xinjiang snow lotus (*Saussurea involucrata* Kar.) and overexpressed. The apigenin content in transgenic Xinjiang snow lotus with the *CHI* gene was 12 times higher than that in the control group, and the total flavonoid content increased 4-fold [73]. The total flavonoid content in transgenic tobacco was 5 times higher than that in the wild type when the jellyfish yalus *CHI* gene was transferred into tobacco [74]. Studies have shown that overexpression of the *CHI* gene can also improve the antioxidant capacity of potato (*Solanum tuberosum* L.) [75]. Flavonoids not only have antibacterial effects, but also are associated with a variety of other biological processes. The results of previous studies on the main synthases in the flavonoid metabolism pathway are relatively clear, but the regulatory network of the flavonoid metabolism pathway has a certain complexity. This network plays an important role in various biological processes, and it is very important to study the key rate-limiting enzyme-encoding genes and homologous gene regulatory networks to elucidate the role of metabolites in the biological processes.

Our experimental results show that the *GbCHI01*, *GbCHI05*, *GbCHI06* and *GbCHI09* genes are all involved in the synthesis of flavonoids, and they are also involved in the defense provided by flavonoids via antibacterial activity in *G*. *barbadense*. The synthesis of flavonoid antibacterial substances is catalyzed and regulated by the *GbCHI01*, *GbCHI05*, *GbCHI06* and *GbCHI09* genes (Figure 7). After the overexpression of the *GbCHI05* gene in 06-146, the flavanones content in the transgenic strains was higher than that in the recipient materials (Appendix A). Flavanones, as intermediate metabolites [76], inhibit Fusarium wilt by feedback regulation and the synthesis of other flavonoid compounds downstream. These results suggest that GbCHI05 gene silencing might rapidly stabilize the metabolic flux of flavonoids through feedback regulation or the rapid activation of other homologous genes to confer defense against Fusarium wilt in *G*. *barbadense*.

Theoretical models of gene functional differentiation after whole-genome duplication involve gene dosage balance [25], subfunctionalization [25,77,78,79] and neofunctionalization [25,45]. Among them, neofunctionalization can occur via numerous different biological processes, such as the acquisition of different transcriptional regulation pathways [80], the subcellular relocalization of proteins [81,82,83] and the acquisition of different protein targets [84,85,86]. However, when the original function of the gene is indispensable, how does neofunctionalization occur? This poorly understood and curious phenomenon is called Ohno’s dilemma [87].

Given that the four genes have the same subcellular localization, the whole-genome duplication-based evolution of the four homologous genes may be explained by the gene dosage balance theory. The four genes cooperate to maintain the stability of flavonoid metabolic flux and synthesize downstream secondary metabolites in an orderly manner, according to the gene dosage balance theory, thus achieving Fusarium wilt resistance. The four homologous genes play an important role in jointly maintaining the dynamic balance of flavonoid synthesis (Figure 8). Through the synergistic regulation of the four genes, the synthetic small-molecular compounds associated with disease resistance are released one at a time, avoiding the one-time release of antibacterial substances, which would affect the response of plants to other stresses and lead to excessive immune damage in the plants. During the process of selection and evolution in plants, the four genes of the *CHI* gene family have specific functions, acting as a cooperative group.

### 3.4. Effect of Endogenous Hormones and ROS on the Neofunctionalization of Homologous Genes

Three phytohormones, MeJA, SA and ethylene (ET), play important regulatory roles in the plant’s defense against pathogens and pests and in the response to abiotic stresses. JA and ET are usually involved in the defense response against necrotizing pathogens. Generally, the MeJA and SA defense pathways are antagonistic to each other, and they act synergistically with the ET defense pathway [88]. Studies have shown that the concentration of JA increases after pathogen infection or tissue damage, and the exogenous application of JA induces the expression of defense-related genes [89]. SA is commonly associated with the defense against biotrophic and semibiotrophic pathogens, as well as the establishment of systemic acquired resistance [90].

Recent studies have shown that JA signaling plays an important role in systemic immunity in *A. thaliana* L. by participating in long-distance information transmission [91]. The rapid accumulation of and increase in JA levels in leaf phloem exudates, and systemwide in leaves, suggests that JA may play a role as a mobile signal involved in pathogen immunity in *A. thaliana* L. [91]. Research on the SA content in Verticillium wilt-resistant *G*. *barbadense* varieties and Verticillium wilt-susceptible upland cotton varieties have has that the SA content in Verticillium wilt-resistant *G*. *barbadense* varieties before and after inoculation was significantly higher than that in Verticillium wilt-susceptible upland cotton varieties [92]. Recent studies have shown that methyl salicylate production is induced during pathogen infection, and this compound acts as a flow inducer in systemic acquired resistance in tobacco [93].

This study showed that silencing the *GbCHI01*, *GbCHI06* and *GbCHI09* genes reduced the accumulation of endogenous MeJA and increased the accumulation of endogenous SA (Figure 9). However, after the *GbCHI05* gene was silenced, the accumulation of endogenous MeJA and endogenous SA was reduced. These four genes might regulate the metabolic flux of flavonoids via the regulation of the levels of endogenous MeJA and SA and simultaneously trigger the immune mechanism of *G*. *barbadense* against Fusarium wilt (Figure 9). After the *GbCHI05* gene was silenced, the changes in the accumulation of endogenous SA were different from those after the silencing of the other genes. These results indicate that the change in the SA content in plants after *GbCHI05* gene silencing may be the key signal or factor that can stabilize the metabolic flux of flavonoid synthesis in plants. In the early stage of resistance to Fusarium wilt, the flavonoid metabolic pathway may be mainly mediated by MeJA, and in the later stage, it may be mainly mediated by SA. In addition, MeJA and SA do not act independently in the resistance of *G*. *barbadense* to Fusarium wilt. The regulatory network associated with MeJA and SA regulation in Fusarium wilt resistance in *G*. *barbadense* and the underlying mechanisms remain to be studied. However, MeJA and SA, as signals, play an important role in the mechanisms by which flavonoid metabolic pathway structural genes regulate Fusarium wilt resistance in *G*. *barbadense* (Figure 9).

ROS are key signaling molecules that enable cells to respond quickly to different stimuli. In plants, ROS play a crucial role in the perception of abiotic stress and biological stress, the integration of different environmental signals, and the activation of stress response networks. In addition, ROS play a key role in the activation and training of plant defense mechanisms [94]. ROS metabolism-related genes also share a certain relationship with fungal pathogenicity. ROS accumulation negatively regulates the pathogenicity of *Fusarium oxysporum* f. sp. *momordicae* [95]. Research data show that QSOX1 in plants is a REDOX sensor that restricts the production of ROS by connecting ROS and active nitrogen signals, and negatively regulates plant immunity [96]. ROS can directly inhibit the growth of pathogens and participate in the disease resistance process as signaling molecules. The rapid production of ROS is an important sign of the activation of the plant’s defense system. After *TaNOX10* gene knockout in *Triticum aestivum* L., the extracellular ROS concentration in *T*. *aestivum* L. was significantly reduced, which confirmed that *TaNOX10* participated in *T*. *aestivum* L. resistance to *Puccinia striiformis* by promoting ROS production. Genetic experiments have confirmed that *TaWRKY19* can specifically bind to W-box elements in the *TaNOX10* promoter and transcriptionally inhibit the expression of the *TaNOX10* gene [97]. When cotton plants sense infection by pathogens such as *Verticillium dahliae*, *GhCaM7* binds to Ca^2+^ as a secondary messenger to enhance the defense response of cotton by activating the JA, ROS and MAPK pathways and changing the cell osmotic potential [98]. In this study, silencing the *GbCHI01*, *GbCHI05*, *GbCHI06* and *GbCHI09* genes directly affected ROS accumulation, indicating that the four *CHI* family genes may directly confer resistance to Fusarium wilt and activate other defense systems of cotton by regulating ROS metabolism (Figure 10).

Gene and genome duplications occur very frequently during plant evolution, and paralogous genes generated by gene duplication during genome evolution provide the basis for genetic redundancy and phenotypic robustness [99]. Gene duplication is a hallmark of plant genome evolution and underlies the genetic interactions that shape phenotypic diversity. Whole-genome duplication events duplicate an entire pathway or network, and the genes in the duplicated pathway or network can evolve new functions through the neofunctionalization and subfunctionalization of genes [100]. Compensation is a major form of homologous gene interaction, with many genes acting as “backup copies” of the original genes. However, how the compensatory relationship changes as allelic variations accumulate is unclear. This makes the result of gene editing less predictable, adding to the difficulty of molecular breeding-based crop improvement efforts [101]. In this study, four genes of the *CHI* gene family related to Fusarium wilt resistance in *G*. *barbadense* were identified, and how they might act cooperatively to confer resistance to Fusarium wilt was explored. The results of this paper reveal a genetic channelization mechanism that can regulate the metabolic flux homeostasis of flavonoids under the mediation of endogenous SA and MeJA by paralogous genes, thereby achieving disease resistance.

Due to the coregulation of paralogous genes in the homeostasis of secondary metabolites due to gene duplication, the antibacterial effects of flavonoids can be exerted through the “one major, three secondary” mode in the *CHI* gene family (one main gene, *GbCHI05*, and three minor genes, *GbCHI01*, *GbCHI06* and *GbCHI09*) (Figure 10). This study demonstrates the ability of plants to maintain normal growth and development in the face of genetic and environmental changes, and provides a theoretical basis for studying the evolutionary patterns of plant homologous genes and for using homologous genes for molecular breeding.

## 4. Materials and Methods

### 4.1. Cloning of GbCHI01, GbCHI05, GbCHI06, and GbCHI09 and Bioinformatics Analysis

The *G*. *barbadense* cultivar used in the test was a resistant cultivar (06-146). Total RNA from cotton was extracted with the Plant Polysaccharide Polyphenol RNA Extraction Kit (TianGen, Beijing, China), and cotton cDNA was synthesized with the Reverse Transcription Kit (Applied Biological Materials, Vancouver, CA, USA). According to the gene sequences of GB_A05G4332 (*GbCHI01*), GB_A13G0219 (*GbCHI05*), GB_D04G0136 (*GbCHI06*) and GB_D13G0209 (*GbCHI09*) [27], Premier 5 software was used to design specific amplification primers for the *GbCHI01*, *GbCHI05, GbCHI06* and *GbCHI09* genes (Appendix A). The amplification program was as follows: 98 °C for 5 min; 35 cycles of 98 °C for 30 s, 57 °C for 30 s, and 72 °C for 1 min for 40 s; and 72 °C for 10 min. The fragments of *G*. *barbadense* GB_A05G4332 (*GbCHI01*), GB_A13G0219 (*GbCHI05*), GB_D04G0136 (*GbCHI06*) and GB_D13G0209 (*GbCHI09*) were amplified and inserted in the pLB-Simple Vector according to the instructions of the Lethal Based Simple Fast Cloning Kit (Tiangen, Beijing, China).

Properties such as the instability index and overall average hydrophilicity were predicted using the online tool ProtParam. PHYRE2 was used to predict the tertiary structure of the GbCHI01, GbCHI05, GbCHI06 and GbCHI09 proteins. Motif Scan software was used to analyze the motifs and domains of *GbCHI* family proteins. Using MEGA 7.0 and ESPript 2.2 software, the amino acid sequences of *GbCHI* family proteins were aligned, and the secondary structure of the CHI protein was described according to the tertiary structure model predicted by the Swiss Model.

### 4.2. Subcellular Localization of the GbCHI01, GbCHI05, GbCHI06 and GbCHI09 Genes

The *GbCHI01*, *GbCHI05*, *GbCHI06*, and *GbCHI09* genes were inserted into subcellular localization vectors with pEarlryGate101 primers as described in Appendix A, and the recombinant plasmids were transformed into *Agrobacterium tumefaciens* GV3101 and transiently expressed in 4-week-old tobacco for 48 h [102]. YFP fluorescence signals were examined for transient expression analysis.

### 4.3. Construction of the VIGS Vector and Fov Infection of GbCHI01, GbCHI05, GbCHI06 and GbCHI09 Gene-Silenced Plants

The *p*TRV2::*GbCHI01*, *p*TRV2::*GbCHI05*, *p*TRV2::*GbCHI06* and *p*TRV2::*GbCHI09* vectors were designed and constructed using infusion technology. The vector primers are shown in Appendix A. Each vector was transformed into *A*. *tumefaciens* GV3101. The *G*. *barbadense* cultivar used in the test was a resistant cultivar (06-146). The gene silencing vectors induced by tobacco mosaic virus were *p*TRV1 and *p*TRV2, and the positive control vector for silencing was *p*TRV2::*CLA1*. 

Cotton seedlings (06-146) were planted in a cotton culture room with a photoperiod of 16 h (light)/8 h (dark), a temperature of 25 °C, and a relative humidity of 60–70% for 8 days. Cotton seedlings with fully expanded cotyledons were injected with the spore solution, and then the injected cotton seedlings were cultured in the dark for 24 h in a cotton culture room [103]. After this cultivation period, the cotton seedlings were planted under suitable conditions. After the above two weeks, injected cotton seedlings were taken from the experimental group and the control group to measure the silencing efficiency. At the three-leaf stage of cotton seedlings with *GbCHI01*, *GbCHI05*, *GbCHI06* and *GbCHI09* gene silencing, the experimental group and the control group were infected with Fusarium wilt [27]. The Fusarium strain used was *Fov* race 7 (a spore solution of approximately 10^7^ spores/mL) [27]. The experiment was repeated 3 times, and at least 30 cotton seedlings were used in each experiment. The statistical analysis of disease symptoms was performed according to the five-level standard [104]. Control plants (*p*TRV2::00) and *GbCHI01*, *GbCHI05*, *GbCHI06* and *GbCHI09* gene-silenced plants (*p*TRV2::*GbCHI01*, *p*TRV2::*GbCHI05*, *p*TRV2::*GbCHI06* and *p*TRV2::*GbCHI09*) were infected with Fusarium wilt. Afterward, the stem segment 3 mm above the cotyledon node was taken for the Fusarium wilt recovery assay. The stems were surface-disinfected with 0.5% sodium hypochlorite (NaClO), 70% ethanol and sterile distilled water and then cultivated on solid potato dextrose agar (PDA) solid for 3 days at 28 °C [105].

### 4.4. Construction of the Overexpression Vector and Fov Infection of GbCHI05-Overexpressing Plants

The pCAMBIA3301::*GbCHI05* vector was designed and constructed using infusion technology. The primers are shown in Appendix A. The pCAMBIA3301::*GbCHI05* vector was transformed into *A*. *tumefaciens* GV3101. The *GbCHI05* gene was overexpressed in Columbia type 0 *A*. *thaliana* and cotton (06-146) by the floral dip method and pollen tube channel method. Positive *A*. *thaliana* seedlings were screened with herbicides at 200 mL/L until T_3_ seeds were obtained. The *GbCHI05* mutants were purchased from the AraShare Technology Service Center. Positive cotton seedlings were screened with herbicides at 90 parts per million (ppm) and qRT–PCR until T_3_ seeds were obtained. 

The seeds of wild type *A. thaliana*, *GbCHI05* mutants and three *GbCHI05* transgenic strains were selected and sterilized with 70% alcohol and 0.7% hypochlorous acid. The seeds were planted in the soil after 3 days of vernalization at 4 °C and then cultured in a greenhouse for 7 days with a photoperiod of 16 h (light)/8 h (dark), a temperature of 20 °C, and a relative humidity of 60–70%. The wild type *A*. *thaliana*, *GbCHI05* mutants and three *GbCHI05* transgenic strains were transplanted into nutrient pots. After 4 weeks, the wild type *A*. *thaliana*, *GbCHI05* mutants and three *GbCHI05* transgenic strains were infected with Fusarium wilt [27], and the phenotype was observed. After infection with Fusarium wilt, control (WT), mutant (*chi05*) and the *GbCHI05-*overexpressing *A*. *thaliana* plants sampled below the first 3 mm of the stem were used for the Fusarium wilt recovery assay [105]. The details of the protocols for identification of Fusarium wilt resistance and the Fusarium wilt recovery assay of cotton overexpression plants are described in the methods above.

### 4.5. Genes Associated with Metabolic Pathways of Flavonoids and Fusarium Wilt

After infection with Fusarium wilt, samples were taken at 0, 2, 4, 8, 12, 24, 48, and 72 h. The samples of the experimental group and the control group were quickly frozen in liquid nitrogen and stored at −80 °C. Metabolic pathway-related genes (*GbC4H*, *GbCHS*, *GbCHI01*, *GbCHI05*, *GbCHI06*, *GbCHI09*, *GbDFR*, *GbF3*′*H*, *GbANR*, *GbFLS* and *GbANS*) and Fusarium wilt-related genes (*GbERF-like* and *Gbar_D03G002290*) were detected. Among them, the *GbERF-like* gene is involved in the resistance to Fusarium wilt in *G*. *barbadense* mediated by SA [30], *Gbar_D03G002290* (*Gh_D03G0209*) is the key gene for resistance to Fusarium wilt [3], and the *Gbar_D03G002290* gene is homologous to the *Gh_D03G0209* gene in upland cotton. Total RNA of cotton was extracted with a plant polysaccharide and polyphenol RNA extraction kit (Tiangen, Beijing, China), and cotton cDNA was synthesized with a reverse transcription kit (Applied Biological Materials, Vancouver, CA, USA). PCR amplification was performed with a real-time quantitative PCR instrument (Applied Biosystems, Foster City, CA, USA) and the SYBR Green dye method (Applied Biological Materials, Vancouver, CA, USA). The reaction conditions were as follows: 3 min at 95 °C; 40 cycles of 15 s at 95 °C and 1 min at 60 °C. The cotton ubiquitin 7 (*UBQ7*) gene was used as an internal reference, and the PCR primer sequences are shown in Appendix A. The biological experiments were repeated 3 times, and the relative gene expression level was analyzed by the 2^−ΔΔCt^ calculation method [106].

### 4.6. Extraction and Quantification of Total Flavonoids

An appropriate amount of rutin was weighed and dissolved in 60% ethanol to prepare the rutin standard solution (0.2 mg/mL). Then, 0.0, 0.5, 1.0, 1.5, 2.0, 2.5, and 3.0 mL of the rutin standard solution (0.2 mg/mL) was placed in 10 mL volumetric flasks, and 60% ethanol was added to a constant volume to prepare the rutin standard reaction mixture. The flavonoid content was determined by an ultraviolet–visible spectrophotometer by measuring the absorption peak at 508 nm. Taking the rutin concentration (mg/mL) as the abscissa and the absorbance at 508 nm as the ordinate, a standard curve was drawn, and the linear regression equation (y = ax + b) was obtained. The absorbance at 508 nm of the reaction mixture of each prepared sample was measured, and the total flavonoid content of the unit sample was calculated according to the linear regression equation.

After infection with Fusarium wilt, the stems and leaves above the base of the cotyledons of cotton seedlings in the experimental and control groups were taken as experimental materials at 0 h and 96 h, respectively. The experimental materials were dried in a constant-temperature electric heating box at 60 °C for 2 h, pulverized in a mortar and passed through a 40-mesh sample sieve, and the powder was collected in a sampling Ziploc bag and marked. A total of 0.05 g of the experimental material was accurately weighed and placed into a 10 mL centrifuge tube. After adding 3 mL of 60% ethanol to the centrifuge tube, the centrifuge tube was heated in a 70 °C constant-temperature water bath for 4 h, and the total flavonoids in the experimental material were fully dissolved in 60% ethanol after heating. After centrifugation, the supernatant solution in the centrifuge tube was placed in a 10 mL volumetric flask [107]. In total 2 mL of the extract solution was placed in a 10 mL volumetric flask, and 0.5 mL of 5% sodium nitrite solution and 0.5 mL of 10% aluminum nitrate solution were added in turn. Then, the mixture was evenly mixed and allowed to stand for 6 min. Then, 5 mL of 4% sodium hydroxide solution and 2 mL of 60% ethanol were added and mixed for 15 min to obtain the sample reaction mixture. Each treatment was repeated three times.

### 4.7. Determination of the Levels of Endogenous Hormones, Flavanone and ROS in Plants

After the experimental group and the control group were infected with Fusarium wilt, samples were taken at 0, 4, 8, 12, 24, 48 and 72 h. The levels of endogenous MeJA and SA in the samples were determined by the enzyme-linked immunosorbent assay (ELISA) in *GbCHI01*, *GbCHI05*, *GbCHI06* and *GbCHI09* gene-silenced plants. The assays were performed according to the instructions for the Plant Methyl Jasmonate ELISA Kit (Jiangsu Jingmei Biotechnology, Yancheng, China) and the Plant Salicylic Acid ELISA Kit (Jiangsu Jingmei Biotechnology, Yancheng, China).

After the experimental group and the control group were infected with Fusarium wilt, samples were taken at 0 and 96 h. The levels of ROS in the samples were determined by enzyme-linked immunosorbent assay (ELISA) in *GbCHI01*, *GbCHI05*, *GbCHI06* and *GbCHI09* gene-silenced plants. The assays were performed according to the instructions for the Plant Reactive Oxygen Species ELISA Kit (Jiangsu Jingmei Biotechnology, Yancheng, China). The levels of flavanone in the samples were determined by enzyme-linked immunosorbent assay (ELISA) in *GbCHI05-*overexpressing cotton plants. The assays were performed according to the instructions for the Flavanone ELISA Kit (Jiangsu Jingmei Biotechnology, Yancheng, China).

## Figures and Tables

**Figure 1 ijms-24-14775-f001:**
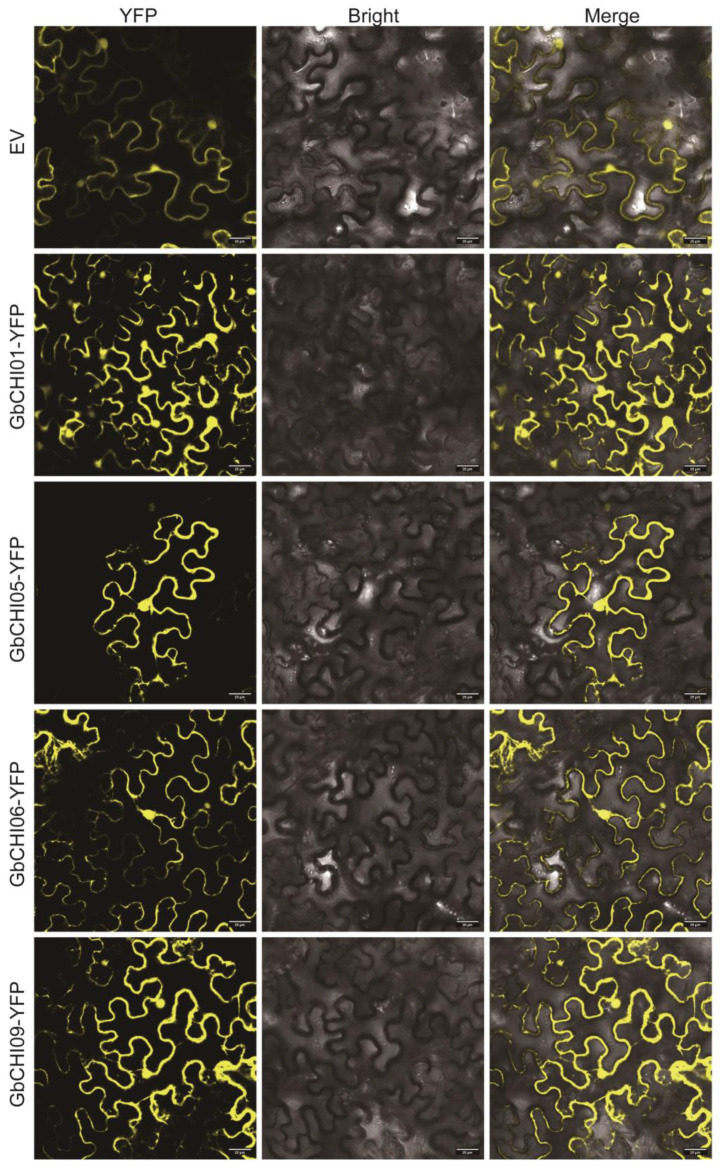
Results of subcellular localization analysis of the *GbCHI* gene from cotton in epidermal cells of *Nicotiana benthamiana* L. Bars, 25 µm.

**Figure 2 ijms-24-14775-f002:**
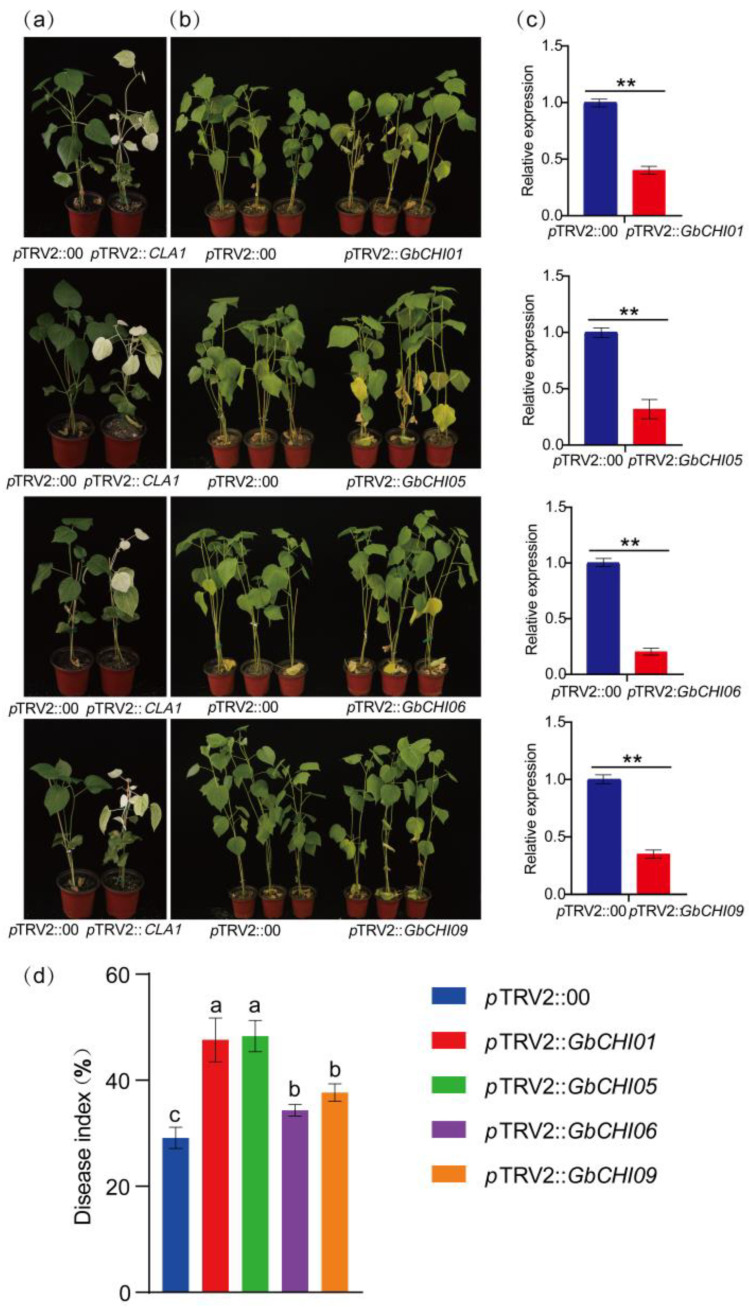
Resistance phenotype of *GbCHI01*, *GbCHI05*, *GbCHI06* and *GbCHI09* gene-silenced cotton plants to Fusarium wilt. (**a**) When *Cloroplastos alterados 1* (*CLA1*) was used as a positive control, the positive control (*p*TRV2::*CLA1*) cotton plants showed an albino phenotype. (**b**) Resistance phenotypes of the control plants (*p*TRV2::00) and the *GbCHI01*, *GbCHI05*, *GbCHI06* and *GbCHI09* gene-silenced plants (*p*TRV2::*GbCHI01*, *p*TRV2::*GbCHI05*, *p*TRV2::*GbCHI06* and *p*TRV2::*GbCHI09*). (**c**) The *GbCHI01*, *GbCHI05*, *GbCHI06* and *GbCHI09* gene silencing efficiency test. In terms of statistical significance, “**” indicates *p* < 0.01. (**d**) The disease index of the phenotypes. a, b, and c above the columns indicate significant differences (*p* < 0.05) according to one-way ANOVA.

**Figure 3 ijms-24-14775-f003:**
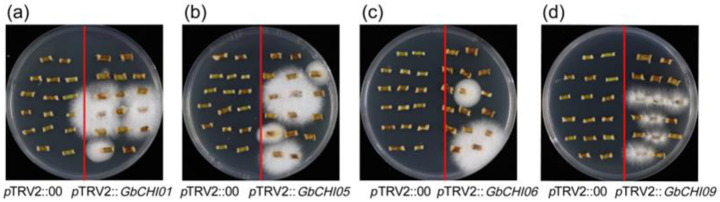
Fusarium wilt recovery assay in silenced plants. The stems 3 mm above the cotyledon node of the control plants (*p*TRV2::00) and the *GbCHI01*, *GbCHI05*, *GbCHI06* and *GbCHI09* gene-silenced plants (*p*TRV2::*GbCHI01*, *p*TRV2::*GbCHI05*, *p*TRV2::*GbCHI06* and *p*TRV2::*GbCHI09*) after infection with Fusarium wilt were taken. (**a**) The stems 3 mm above the cotyledon node of the control plants and the *GbCHI01* gene-silenced plants. (**b**) The stems 3 mm above the cotyledon node of the control plants and the *GbCHI05* gene-silenced plants after infection with Fusarium wilt were taken. (**c**) The stems 3 mm above the cotyledon node of the control plants and the *GbCHI06* after infection with Fusarium wilt were taken. (**d**) The stems 3 mm above the cotyledon node of the control plants and the *GbCHI09* gene-silenced plants after infection with Fusarium wilt were taken.

**Figure 4 ijms-24-14775-f004:**
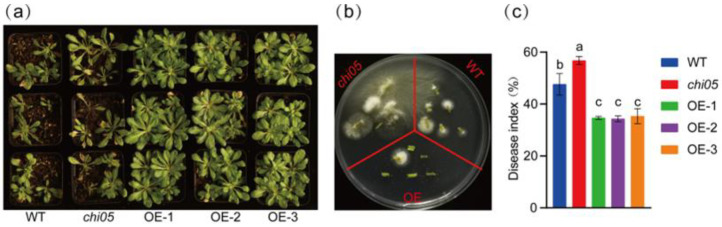
(**a**) Phenotypic identification of *GbCHI05* gene overexpression in *Arabidopsis thaliana* for resistance to Fusarium wilt. WT: wild type. *chi05*: a *GbCHI05* gene mutant. OE-1, OE-2 and OE-3: three *GbCHI05* transgenic lines. (**b**) Fusarium wilt recovery assay in *A. thaliana GbCHI05-*overexpressing plants. After infection with Fusarium wilt, control (WT), mutant (*chi05*) and *GbCHI05*-overexpressing *A. thaliana* plants were grown, and the 3 mm stem segment of the hypocotyledonary axis was examined. (**c**) The disease index of the phenotypes. a, b, and c above the columns indicate significant differences (*p* < 0.05) according to one-way ANOVA.

**Figure 5 ijms-24-14775-f005:**
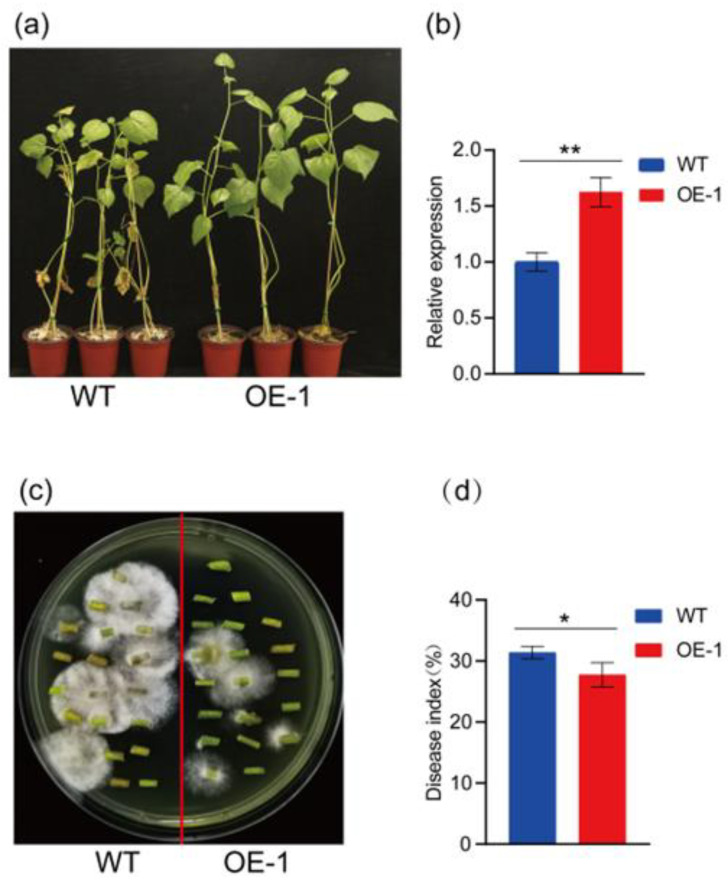
Phenotypic identification of *GbCHI05* gene overexpression in cotton against Fusarium wilt and Fusarium wilt recovery assay. (**a**) Resistance phenotype analysis under *GbCHI05* gene overexpression in cotton. (**b**) Determination of *GbCHI05* expression level in cotton plants overexpressing this gene. (**c**) Fusarium wilt recovery assay. WT: 06-146 (*Gossypium barbadense* L.). OE-1: transgenic lines of the *GbCHI05* gene in T_3_ seedlings. The receptor of the transgenic material is 06-146. In terms of statistical significance, “**” indicates *p* < 0.01. (**d**) The disease index of the phenotypes. In terms of statistical significance, “*” indicates *p* < 0.05.

**Figure 6 ijms-24-14775-f006:**
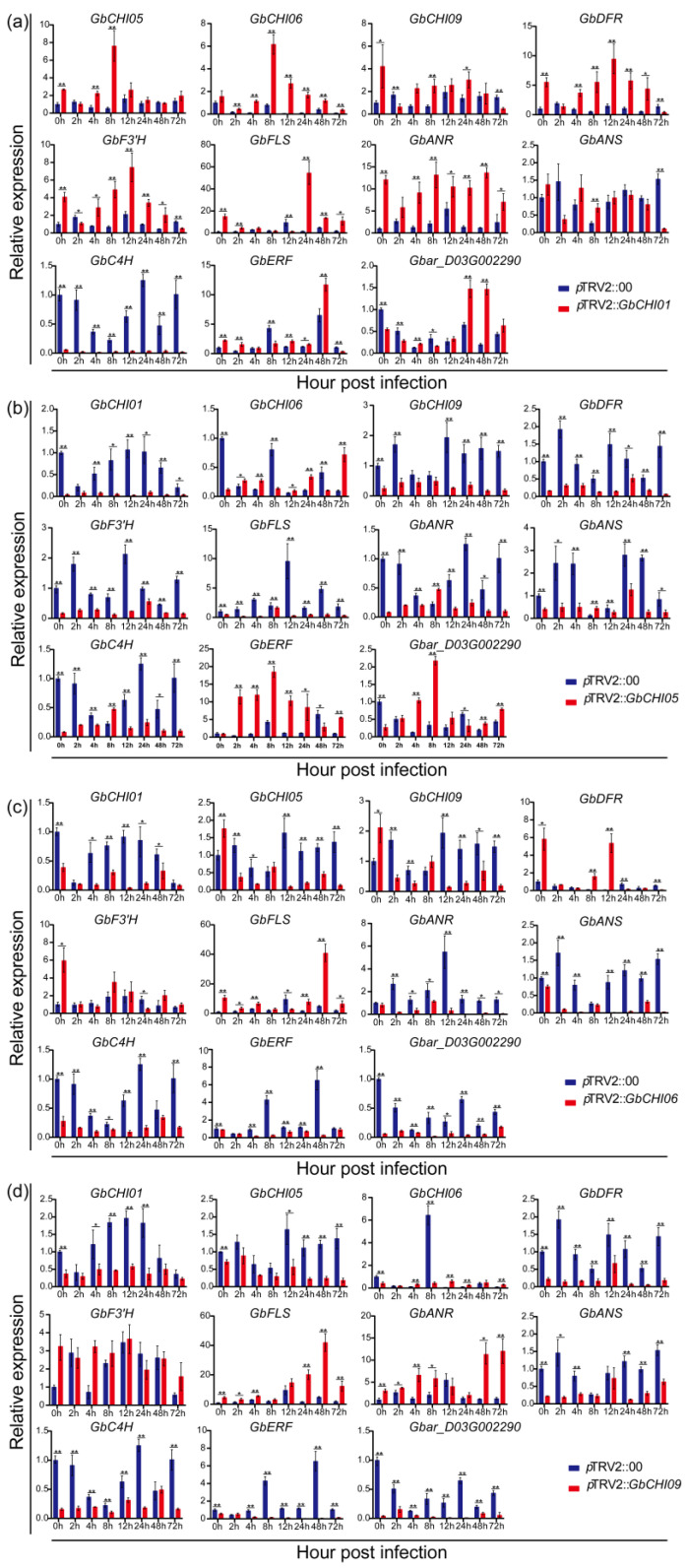
Detection of genes involved in the flavonoid pathway and genes related to Fusarium wilt in control plants (*p*TRV2::00) and *GbCHI01*, *GbCHI05*, *GbCHI06* and *GbCHI09* gene-silenced plants (*p*TRV2::*GbCHI01*, *p*TRV2::*GbCHI05*, *p*TRV2::*GbCHI06* and *p*TRV2::*GbCHI09*). Samples were taken at 0, 2, 4, 8, 12, 24, 48 and 72 h; (**a**) *GbCHI01* gene-silenced plants; (**b**) *GbCHI05* gene-silenced plants; (**c**) GbCHI06 gene-silenced plants; (**d**) *GbCHI09* gene-silenced plants. In terms of statistical significance, “*” is used to indicate *p* < 0.05, and “**” indicates *p* < 0.01.

**Figure 7 ijms-24-14775-f007:**
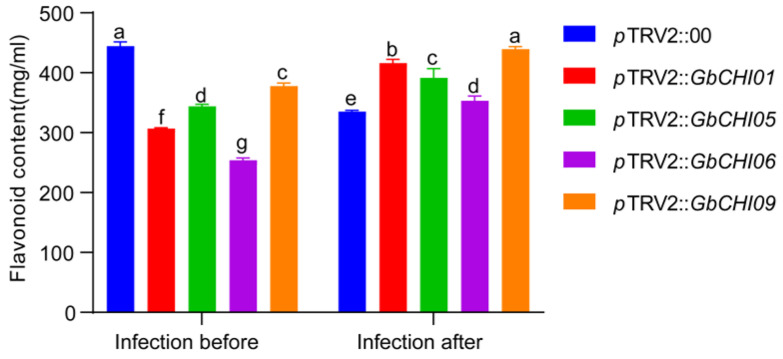
Establishment of a rutin standard curve for measuring flavonoid content in cotton. Flavonoid content of the control plants (*p*TRV2::00) and the *GbCHI01*, *GbCHI05*, *GbCHI06* and *GbCHI09* gene-silenced plants (*p*TRV2::*GbCHI01*, *p*TRV2::*GbCHI05*, *p*TRV2::*GbCHI06* and *p*TRV2::*GbCHI09*) before and after infection with the causal agent of Fusarium wilt. a, b, c, d, e, f and g above the columns indicate significant differences (*p* < 0.05) according to one-way ANOVA.

**Figure 8 ijms-24-14775-f008:**
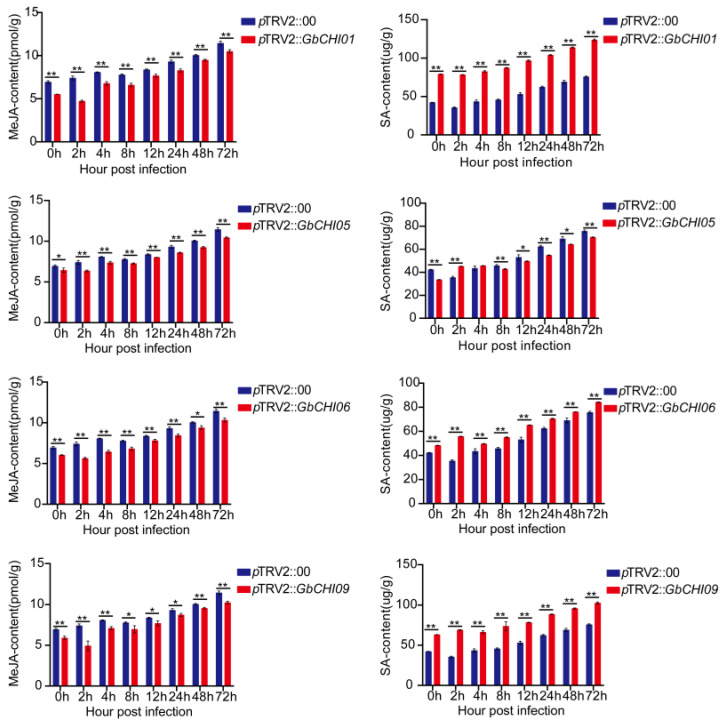
Endogenous hormone content (MeJA and SA) after *Fov* infection. *p*TRV2::00: control plants. *p*TRV2::*GbCHI01*, *p*TRV2::*GbCHI05*, *p*TRV2::*GbCHI06* and *p*TRV2::*GbCHI09*: *GbCHI01*, *GbCHI05*, *GbCHI06* and *GbCHI09* gene-silenced plants. Samples were taken at 0, 2, 4, 8, 12, 24, 48 and 72 h; “*” indicates *p* < 0.05, and “**” indicates *p* < 0.01.

**Figure 9 ijms-24-14775-f009:**
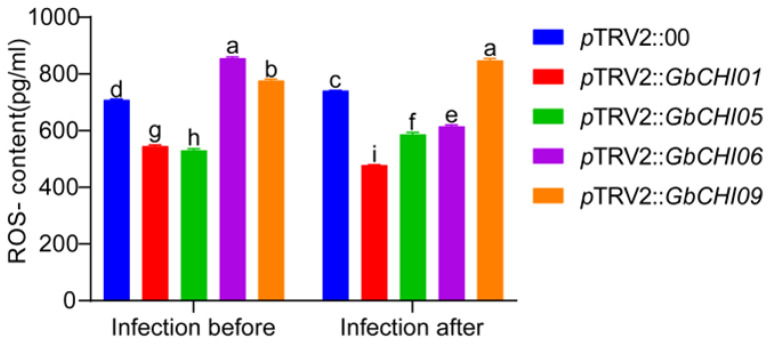
ROS contents after *Fov* infection. ROS contents of the control plants (*p*TRV2::00) and the *GbCHI01*, *GbCHI05*, *GbCHI06* and *GbCHI09* gene-silenced plants (*p*TRV2::*GbCHI01*, *p*TRV2::*GbCHI05*, *p*TRV2::*GbCHI06* and *p*TRV2::*GbCHI09*) before and after infection with the causal agent of Fusarium wilt. a, b, c, d, e, f, g, h and i above the columns indicate significant differences (*p* < 0.05) according to one-way ANOVA.

**Figure 10 ijms-24-14775-f010:**
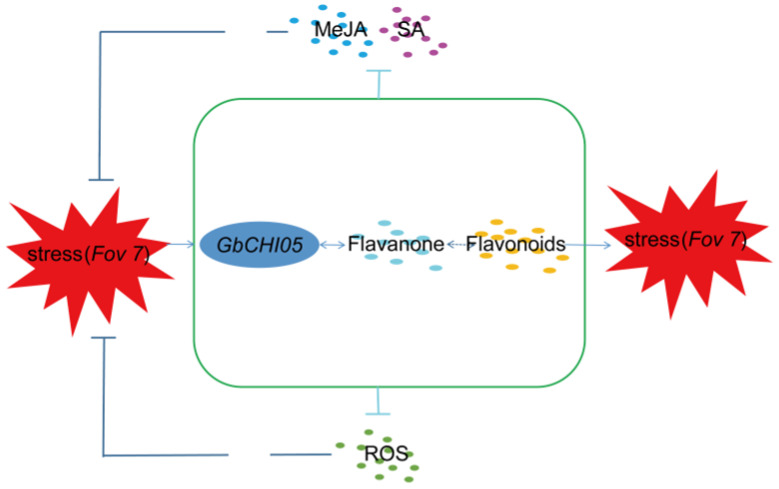
The “one major, three secondary” mode in the *CHI* gene family.

## Data Availability

All the data generated or analyzed during this study are included in this article and its Appendix A.

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
