# Peer review of "Genetic Channelization Mechanism of Four Chalcone Isomerase Homologous Genes for Synergistic Resistance to Fusarium wilt in Gossypium barbadense L."

_ijms, 2023, doi:10.3390/ijms241914775_

Round 1
Reviewer 1 Report
COMMENTS TO THE AUTHOR:
This ARTICLE entitled "Genetic Channelization Mechanism of Four Chalcone Isomerase Homologous Genes for Synergistic Resistance to Fusarium wilt in Sea Island Cotton" by Zu et al investigated the function of GbCHIs in disease resistance systematically coming from a large amount of works. It’ s well written and merit to be published after addressing some comments.
Minor concerns:
1. Figure 1, to confirm the membrane localization, plasmolysis experiment is more convincing.
2. Figure 2, the disease index showed be calculated along with the resistance phenotypes.
3. The figure icons should be more cited in the results content.
4. Figure 3, we hardly observed the Fusarium wilt biomass in the control plants (TRV::00), however, it is unreasonable.
5. Figure 4, Lacking statistical data.
6. Why the authors only overexpressed GbCHI05 for the functional analysis in Arabidopsis and cotton?
7. Figure 5, it looks like other types of fungi has developed, in addition, Figure 4 and 5 should be combined.
8. Figure 6, which the generation was used, and whether the overexpressed lines are homozygous were not descripted.
9. Line143-144, Line245-250, moved to the discussion section.
The quality of English can be improved.
Author Response
Point 1: Figure 1, to confirm the membrane localization, plasmolysis experiment is more convincing.
Response 1: We are very sorry for our unclear displaying. Thank you for your suggestions, which are very important and have important guiding significance for my thesis writing and scientific research work. Our current experimental conditions are not mature enough, the experimental system is not perfect. In the later stage, if the conditions are ripe, it will be improved in the future similar studies. The text of “ The subcellular localization of the GbCHI01, GbCHI05, GbCHI06 and GbCHI09 proteins in the nucleus and cell membrane was evident. ” were corrected as “ The GbCHI01, GbCHI05, GbCHI06 and GbCHI09 proteins may be localized to the nucleus and cell membrane. ”. The text of “ In this study, we found that the nucleus and cell membrane were the main functional sites of the CHI protein, ” were corrected as “ In this study, we found that the nucleus and cell membrane might be the main functional sites of the CHI protein, ”.
Point 2: Figure 2, the disease index showed be calculated along with the resistance phenotypes.
Response 2: We are very sorry for our unclear displaying. We have revised this part according to the Reviewer’s suggestion. We added the content of the disease index and the text of “ (d) The disease index of the phenotypes. a, b, and c above the columns indicate significant differences (P<0.05) according to one-way ANOVA. ” in Figure 2.
Point 3: The figure icons should be more cited in the results content.
Response 3: We are very sorry for our incorrect displaying. We have revised this part according to the Reviewer’s suggestion. The figure icons are more frequently cited in the results and discussion sections, as detailed in the manuscript.
Point 4: Figure 3, we hardly observed the Fusarium wilt biomass in the control plants (TRV::00), however, it is unreasonable.
Response 4: We are very sorry for our unclear displaying. Because the stem segment materials used in the Fusarium wilt recovery assay were different. In Fov infection of gene-silenced plants and Fov infection of GbCHI05-overexpressing plants were infected with Fusarium wilt, the results were different. The method of gene-silenced plants is the text of ” The stem 3 mm above the cotyledon node was taken for the Fusarium wilt recovery assay. ”. The method of GbCHI05-overexpressing plants is the text of ” After infection with Fusarium wilt, control plants (WT), mutants (chi05) and the GbCHI05-overexpressing Arabidopsis thaliana sampled below the first 3 mm of the stem were used for the Fusarium wilt recovery assay [103]. The details of the protocols for identification of Fusarium wilt resistance and the Fusarium wilt recovery assay of cotton overexpressing plants are located in the abovementioned methods. ”. In addition, the situation in Figure 2f in the reference " Liu S, Zhang X, Xiao S, et al. A single‐nucleotide mutation in a GLUTAMATE RECEPTOR‐LIKE gene confers resistance to Fusarium Wilt in Gossypium hirsutum[J]. Advanced Science, 2021, 8(7): 2002723. " is similar to our manuscript, and we have carried out three repetitions with the same results.
Point 5: Figure 4, Lacking statistical data.
Response 5: We are very sorry for our unclear displaying. We have revised this part according to the Reviewer’s suggestion. We added the statistical data and the text of ” (c) The disease index of the phenotypes. a, b, and c above the columns indicate significant differences (P<0.05) according to one-way ANOVA. ” in Figure 4.
Point 6: Why the authors only overexpressed GbCHI05 for the functional analysis in Arabidopsis and cotton?
Response 6: We are very sorry for our unclear displaying. After GbCHI05 gene-silenced, the disease index of GbCHI05 gene-silenced plants in cotton was the highest, and the disease was the most serious. The changes of endogenous SA accumulation after GbCHI05 gene-silenced were different from those after other gene-silenced. These results suggest that GbCHI05 gene may be a key signal or factor in stabilizing the anabolic flux of flavonoids in cotton. Therefore, we only overexpressed GbCHI05 and further performed functional analysis in Arabidopsis and cotton.
Point 7: Figure 5, it looks like other types of fungi has developed, in addition, Figure 4 and 5 should be combined.
Response 7: We are very sorry for our incorrect displaying. We have revised this part according to the Reviewer’s suggestion. Due to the stems were surface disinfected with 0.5% sodium hypochlorite (NaClO), 70% ethanol and sterile distilled water and then cultivated on solid potato dextrose agar (PDA) solid for 3 days at 28 ℃. Chlorophyll from the stem have stained the Fusarium oxysporum f. sp. vasinfectum green. At the same time, we reconstructed the images with the photos of the repeated experiment. We combined Figure 4 and Figure 5 to form the new Figure 4. We also have added the text of “Statistical analysis of the disease index showed that the disease index of the wild type (WT) and chi05 mutant were significantly higher than that of the GbCHI05-overexpressing strains.” in result.
Point 8: Figure 6, which the generation was used, and whether the overexpressed lines are homozygous were not descripted.
Response 8: We are very sorry for our incorrect displaying. We have revised this part according to the Reviewer’s suggestion. We added the text of “ After the overexpression of the GbCHI05 gene in T3 06-146 seeds, the 06-146 (WT) were cultured in the greenhouse. ” in result. We also added the text of “ in T3 06-146 seedlings ” in the note to Figure 2
Point 9: Line143-144, Line245-250, moved to the discussion section.
Response 9: We are very sorry for our incorrect displaying. We have revised this part according to the Reviewer’s suggestion. We moved those two sections to the materials and discussion section, as detailed in the manuscript.
Reviewer 2 Report
In the manuscript entitled "Genetic Channelization Mechanism of Four Chalcone Isomerase Homologous Genes for Synergistic Resistance to Fusarium Wilt in Sea Island Cotton," Zu et al. performed a comprehensive analysis of CHI genes in cotton, including resistance to Fusarium wilt, expression of metabolic pathway-related genes, metabolite content, endogenous hormone content, reactive oxygen species (ROS) content, and subcellular localization. They found that CHI may affect SA and MeJA, subsequently influencing plant disease resistance. This study provided new targets for cotton breeding. I have some suggestions for this manuscript before it can be accepted:
1. The English writing should be improved. There are some typos and grammatical errors in this draft MS. For example, in the abstract section, in line 19, "Fusarium wilt infection" should be changed to "Fusarium oxysporum f. sp. vasinfectum infection." In line 20, a full stop is missing before the word "However." In line 29, “it” should be "Our study." Line 25 mentions "the result" but in line 26, "the result" is mentioned again. The sentence in lines 105-109 is highly similar to those in the abstract section. The authors need to rephrase them. In Figure 8 and 10, "Befour" should be corrected to "Before."
2. In the phrase "Fusarium wilt," "Fusarium" should not be italicized. Only when it represents the Latin name of fungi, it should be italicized.
3. A cell membrane marker should be included in Figure 1. Alternatively, the authors can perform cytoplasmic and cell wall separation experiments to confirm the localization of these genes in the cell membrane. Moreover, DAPI staining should also be added to indicate nuclear localization.
4. The disease index for the VIGS plants should be added in Figure 2. The details for the phenotyping should also be elucidated in the context. In line 175, the standard deviation (SD) value for DI should also be displayed.
5. All the "P" values indicating statistical significance should be italicized.
6. The authors should provide an explanation in the context regarding why they selected GbHI105. Additionally, Figure 4 and Figure 5 can be merged into one figure. Furthermore, the disease index (DI) for Figure 4 should be included.
7. In Figure 6, the authors should provide information regarding whether the WT is Hirsutum or Babardense. It is important to clarify which type of cotton was used as the wild type.
8.The different expression patterns of GbDFR, GbF3'H, GbFLS, GbANR, and GbANS in different VIGS plants should be and discussed.
The English writing need to be improved.
Author Response
Point 1: The English writing should be improved. There are some typos and grammatical errors in this draft MS. For example, in the abstract section, in line 19, "Fusarium wilt infection" should be changed to "Fusarium oxysporum f. sp. vasinfectum infection." In line 20, a full stop is missing before the word "However." In line 29, “it” should be "Our study." Line 25 mentions "the result" but in line 26, "the result" is mentioned again. The sentence in lines 105-109 is highly similar to those in the abstract section. The authors need to rephrase them. In Figure 8 and 10, "Befour" should be corrected to "Before."
Response 1: We are very sorry for our incorrect displaying. We have re-written this part according to the Reviewer’s suggestion. The text of “ Fusarium wilt infection ” were corrected as “ Fusarium oxysporum f. sp. vasinfectum infection ”. The text of “ it ” were added a full stop before the word "However". The text of “ it ” were corrected as “ Our study ”. The text of “ the result ” were corrected as “ those results ”. The text of “In this study, we investigated the molecular specificity, Fusarium wilt resistance, metabolic pathway-related gene expression, metabolite content changes, endogenous hormone content changes, reactive oxygen species(ROS) content and subcellular localization of the four paralogous genes of the CHI gene family in G. barbadense and found that the four paralogous genes function synergistically. The results of this paper reveal a genetic channelization mechanism that can regulate the metabolic flux homeostasis of flavonoids under the mediation of endogenous salicylic acid (SA) and methyl jasmonate (MeJA) via four CHI paralogous genes, thereby achieving disease resistance. ” were corrected as “ In this study, we identified the molecular specificity and function of four paralogous genes in G. barbadense and and found that the four paralogous genes function synergistically. This study revealed that homologous genes are synergically regulated in the homeostasis of secondary metabolites due to gene duplication, and the antibacterial effect of flavonoids can be exerted through the "1 major 3" pattern of GbCHI gene family (1 major gene GbCHI05, 3 minor genes GbCHI01, GbCHI06 and GbCHI09).”. The text of “ Fusarium wilt infection ” were corrected as “ Fusarium oxysporum f. sp. vasinfectum infection ”. The text of “ Fusarium wilt infection ” were corrected as “ Fusarium oxysporum f. sp. vasinfectum infection ”. In Figures, "Befour" have be corrected to "Before."
Point 2: In the phrase "Fusarium wilt," "Fusarium" should not be italicized. Only when it represents the Latin name of fungi, it should be italicized.
Response 2: We are very sorry for our incorrect displaying. We have revised this part according to the Reviewer’s suggestion. The text of “ Fusarium ” were corrected as “Fusarium ” except when it stands for the Latin name of the fungus.
Point 3: A cell membrane marker should be included in Figure 1. Alternatively, the authors can perform cytoplasmic and cell wall separation experiments to confirm the localization of these genes in the cell membrane. Moreover, DAPI staining should also be added to indicate nuclear localization.
Response 3: We are very sorry for our unclear displaying. Thank you for your suggestions, which are very important and have important guiding significance for my thesis writing and scientific research work. In the later stage, if the conditions are ripe, it will be improved in the future similar studies.
Point 4: The disease index for the VIGS plants should be added in Figure 2. The details for the phenotyping should also be elucidated in the context. In line 175, the standard deviation (SD) value for DI should also be displayed.
Response 4: We are very sorry for our unclear displaying. We have revised this part according to the Reviewer’s suggestion. We have added the content of the disease index and ”(d) The disease index of the phenotypes. a, b, and c above the columns indicate significant differences (P<0.05) according to one-way ANOVA.” in Figure 2. And the standard deviation (SD) value for DI was also displayed in Figure 2.
Point 5: All the "P" values indicating statistical significance should be italicized.
Response 5: We are very sorry for our incorrect displaying. We have revised this part according to the Reviewer’s suggestion. The text of “ P ” were corrected as italics.
Point 6: The authors should provide an explanation in the context regarding why they selected GbHI105. Additionally, Figure 4 and Figure 5 can be merged into one figure. Furthermore, the disease index (DI) for Figure 4 should be included.
Response 6: We are very sorry for our unclear displaying. We have revised this part according to the Reviewer’s suggestion. After GbCHI05 gene-silenced, the disease index of GbCHI05 gene-silenced plants in cotton was the highest, and the disease was the most serious. The changes of endogenous SA accumulation after GbCHI05 gene-silenced were different from those after other gene-silenced. These results suggest that GbCHI05 gene may be a key signal or factor in stabilizing the anabolic flux of flavonoids in cotton. We combined Figure 4 and Figure 5 to form the new Figure 4. We have added the text of “ Statistical analysis of the disease index showed that the disease index of the wild type (WT) and chi05 mutant were significantly higher than that of the GbCHI05-overexpressing strains.” in result. We added the statistical data and the text of ” (c) The disease index of the phenotypes. a, b, and c above the columns indicate significant differences (P<0.05) according to one-way ANOVA.” in Figure 4.
Point 7: In Figure 6, the authors should provide information regarding whether the WT is Hirsutum or Babardense. It is important to clarify which type of cotton was used as the wild type.
Response 7: We are very sorry for our incorrect displaying. We have revised this part according to the Reviewer’s suggestion. We have marked in Figure 6 that WT is Gossypium barbadense L. We also added the text of “ (d) The disease index of the phenotypes. In terms of statistical significance, "*" indicates P < 0.05. ” in Figure 6.
Point 8: The different expression patterns of GbDFR, GbF3'H, GbFLS, GbANR, and GbANS in different VIGS plants should be and discussed.
Response 8: We are very sorry for our incorrect displaying. We have revised this part according to the Reviewer’s suggestion. We have added the text of “ The expression patterns of GbDFR, GbF3’H and GbFLS were the same when GbCHI01, GbCHI06 and GbCHI09 were silenced, respectively. The expression patterns of GbANS were different in other silenced plants when GbCHI05 was silenced. In GbCHI01 gene-silenced plants, the expression pattern of GbANR is also different in other gene-silenced plants. ” in discussion.
Reviewer 3 Report
The paper under review’’ Genetic Channelization Mechanism of Four Chalcone Isomerase Homologous Genes for Synergistic Resistance to Fusarium wilt in Sea Island Cotton’’ investigates resistance to Fusarium wilt, expression of metabolic pathway-related genes, metabolite content, endogenous hormone content, reactive oxygen species (ROS) content, and subcellular localization of four paralogous CHI family genes. This study introduces an important topic regarding the impact of gene duplication events on plant evolution and functional diversification. Further, it explores the potential for neofunctionalization and subfunctionalization within duplicated genes, specifically focusing on the CHI gene family's role in Fusarium wilt resistance in cotton. While the current study presents intriguing insights, there are several aspects that warrant further clarification and improvement.
Suggestions for Improvement:
- Clarity and Flow: The last sentence of the introduction is quite lengthy and could be broken down into smaller sentences for better clarity.
- Background Information: Provide a brief explanation of what CHI genes are and their role in plant defense mechanisms. This would help readers who might not be familiar with the term.
- Research Gap Clarification: Elaborate on the specific aspects of neofunctionalization differences within the CHI gene family that have not been adequately explored in previous literature.
- Implications and Significance: Explicitly state the potential implications of the discovered genetic channelization mechanism and how it contributes to disease resistance. Additionally, elaborate on the theoretical basis for studying evolutionary patterns and using homologous genes for molecular breeding.
Overall Recommendation:
Given the promising insights into the genetic mechanisms behind plant resistance and evolution, I recommend accepting the Manuscript with the suggested minor revisions. Addressing the aforementioned points will enhance the clarity and comprehensibility of the manuscript, making it a valuable contribution to the field.
Author Response
Point 1: I Clarity and Flow: The last sentence of the introduction is quite lengthy and could be broken down into smaller sentences for better clarity.
Response 1: We are very sorry for our unclear description. We have re-written this part according to the Reviewer’s suggestion. The text of “This study provides evidence for genetic redundancy and phenotypic robustness and offers a strategy for ensuring plant stability to maintain normal growth and development in the face of genetic and environmental changes.” were corrected as “This study provides evidence for genetic redundancy and phenotypic robustness. In addition, this study offers a strategy for ensuring plant stability to maintain normal growth, and development in the face of genetic and environmental changes.”.
Point 2: Background Information: Provide a brief explanation of what CHI genes are and their role in plant defense mechanisms. This would help readers who might not be familiar with the term.
Response 2: We are very sorry for our unclear description. We have revised this part according to the Reviewer’s suggestion. We added the text of “ However, most miRNAs targeting CHI genes are associated with cotton immunity [11]. In addition,, GbCHI protein inhibits spore germination and mycelium growth of Verticillium dahliae. [12]. ” in introduction.
Point 3: Research Gap Clarification: Elaborate on the specific aspects of neofunctionalization differences within the CHI gene family that have not been adequately explored in previous literature.
Response 3: We are very sorry for our unclear displaying. Thank you for your suggestions, which are very important and have important guiding significance for my thesis writing and scientific research work. Previous studies have not found any research on the new functional differentiation of CHI gene. This paper discusses and conducts a preliminary study on the new functional differentiation of CHI gene for the first time, and the specific aspects of the new functional differentiation of CHI gene will be studied in the future.
Point 4: Implications and Significance: Explicitly state the potential implications of the discovered genetic channelization mechanism and how it contributes to disease resistance. Additionally, elaborate on the theoretical basis for studying evolutionary patterns and using homologous genes for molecular breeding.
Response 4: We are very sorry for our unclear displaying. Thank you for your suggestions, which are very important and have important guiding significance for my thesis writing and scientific research work. In our study, given that the four genes have the same subcellular localization, the whole-genome duplication-based evolution of the four homologous genes may be explained by the gene dosage balance theory. The four genes cooperate to maintain the stability of flavonoid metabolic flux and synthesize downstream secondary metabolites in an orderly manner according to the gene dosage balance theory, thus achieving Fusarium wilt resistance. The four homologous genes play an important role in jointly maintaining the dynamic balance of flavonoid synthesis (Figure 8). Through the synergistic regulation of the four genes, the synthetic small-molecular compounds associated with disease resistance are released one at a time, avoiding one-time release of antibacterial substances, which would affect the response of plants to other stresses and lead to excessive immune damage in plants. During the process of selection and evolution in plants, the four genes of the CHI gene family have specific functions, acting as a cooperative group.
Reviewer 4 Report
Well-written and nicely investigated paper. Please consider indicating the species (Gossypium barbadense) of Sea Island types. Some claims are made regarding novelty of investigation specifically for Sea Island types when other investigations include G. barbadense of other types. Gb genes could denote G. barbadense but the species should be identified early (as in your #25 Zu et al., 2019 reference). Discussion is excellent but could be more concise.
Author Response
Point 1: Well-written and nicely investigated paper. Please consider indicating the species (Gossypium barbadense) of Sea Island types. Some claims are made regarding novelty of investigation specifically for Sea Island types when other investigations include G. barbadense of other types. Gb genes could denote G. barbadense but the species should be identified early (as in your #25 Zu et al., 2019 reference).
Response 1: We are very sorry for our unclear description. We have revised this part according to the Reviewer’s suggestion. The text of “ Sea Island cotton” were corrected as “Gossypium barbadense L.”.
Point 2: Discussion is excellent but could be more concise.
Response 2: We are very sorry for our unclear description. We have revised this part according to the Reviewer’s suggestion. We deleted the text of “ This may be related to the fact that the GbCHI01, GbCHI05, GbCHI06 and GbCHI09 genes have the same function in plants. However, ”, the text of “ The nucleus and cell membrane are important sites for plant genetic material storage and external signal transduction and recognition, as well as the main sites of localization of genetic material and genes and enzymes required for signal transduction and recognition.” and the text of “ The smallest change in flavonoid content after Fov infection was observed in plants subjected to GbCHI05 gene silencing, followed by plants subjected to GbCHI09 gene silencing, in which the change was less than that in the other two lines (GbCHI01 and GbCHI06) (Figure 8). ” in discussion.
Round 2
Reviewer 1 Report
All my concerns have been well answered.
The English is well.
Author Response
Dear reviewer, thank you for your valuable comments.
Reviewer 2 Report
The authors addressed all my concerns in the revised MS, which now can be accepted in IJMS.
Author Response

(The authors gave the same response as above.)
